# Demystifying the Underappreciated Long-Tail Problems in Large Vision Language Models

## Abstract

Recently, Large Vision-Language Models (LVLMs) have made significant progress, seamlessly integrating the visual comprehension capabilities of vision encoders with the language generation strengths of language models (LMs). Despite the success of LVLMs, the training or aligning data of LVLMs suffers from the *Long-Tail (LT)* problems, which is a special type of data with highly imbalanced distributions, and a large number of tail (minority) instances. A significant amount of research has focused on mitigating LT through data adjustment or network structure reorganization, however, efforts targeting generative LVLMs remain limited. In this paper, we present an in-depth analysis of the LT issues persisting in LVLMs' training data and build a distribution of four perspectives, addressing both visual and language aspects. To mitigate the aforementioned challenges, we propose an **A**daptive **D**ata **R**efinement Framework (**ADR**), which consists of two stages: **D**ata **R**ebalancing (DR) and **D**ata **S**ynthesis (DS). In the DR stage, we adaptively rebalance the redundant data based on entity distributions, while in the DS stage, we leverage the latent representations of scarce images to adaptively supplement the underrepresented portions. To validate the effectiveness of our approach, we conduct experiments on a series of comprehensive benchmarks, including the GPT-assisted evaluations to assess the overall performance variations introduced by our method. Through comprehensive evaluations, ADR effectively mitigates the long-tail problem in the training data, improving the average performance of LLaVA 1.5 relatively by **2.62%** across 10 benchmarks, without increasing the training data volume. Our code and data will be publicly released.

## 1 Introduction

Large Vision-Language Models (LVLMs) have become pivotal at the intersection of computer vision and natural language processing, enabling breakthroughs in bridging the gap between language and vision. These models facilitate a wide range of applications by generating contextually relevant textual descriptions from visual inputs. Recent advancements in LVLMs (Bai et al., 2023; Chen et al., 2023a; Dai et al., 2024; Zhang et al., 2023a; Dong et al., 2024; Chen et al., 2023c; Liu et al., 2023a; 2024a; Zhu et al., 2023; Ye et al., 2023b; Abdin et al., 2024) have significantly advanced general-purpose foundation models, elevating them to unprecedented levels. This flourishing progress within the research community marks a significant step towards Artificial General Intelligence (AGI).

However, the training data of LVLMs are suffering from the problem of *Long-Tail (LT)* (Parashar et al., 2024). The "Long Tail" term in computer vision refers to the fact that the training datasets nowadays are often of large scale and present highly imbalanced distributions, and they have a large number of tail (minority) classes. Although LT is common in real-world distribution and cannot be seen as a problem, recent research (Zhang et al., 2024; 2023b; Fu et al., 2022; Yang et al., 2022) have found that balancing the LT data could bring positive effects. Several works (Shao et al., 2023; Wang et al., 2022; Zhu et al., 2024; Liu et al., 2024b; Lee et al., 2024) have been made to mitigate LT by either balancing the redundant data or re-designing the network structure.

Despite the efforts, these approaches are not yet fully satisfying in terms of eliminating the LT problem within the training data of LVLMs. Some existing methods focus on traditional models (e.g., CLIP (Radford et al., 2021)) or tasks (e.g., image classification), while others prioritize data efficiency, aiming to achieve comparable performance with fewer data. However, none of these

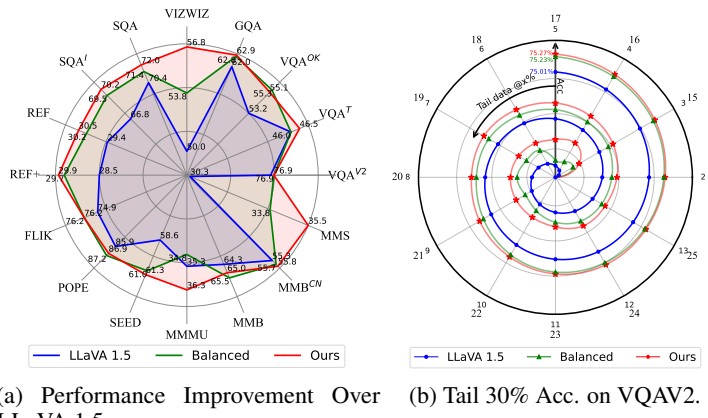

(a) Performance Improvement Over LLaVA 1.5.

(b) Tail 30% Acc. on VQAV2.

Figure 1: Performance on comprehensive benchmarks of LLaVA 1.5 before and after addressing the long-tail problem. Our method surpasses the baseline over **16/16** benchmarks and improves the performance of tail **30%** concepts.

approaches have significantly enhanced the overall performance of generative LVLMs by addressing LT or optimizing the training data.

To address the above issues, we delve into the distribution of the training data, analyzing it from four perspectives based on both language and visual features, and introduce An **A**daptive **D**ata **R**efinement Framework (**ADR**), a novel data calibration strategy designed to effectively counter LT and can be easily integrated into any open-source LVLMs' training data such as LLaVA (Liu et al., 2023a), ShareGPT-4V (Chen et al., 2023b) ALLaVA (Chen et al., 2024b) or Mini-GPT4V (Zhu et al., 2023). ADR addresses the long-tail (LT) problem by both filtering redundant data and supplementing scarce data through two key stages: **D**ata **R**ebalancing Stage (DR) and **D**ata **S**ynthesis Stage (DS). The DR stage rebalances the overrepresented head portion of the data by filtering out low-quality or redundant instances based on entity distributions, thereby mitigating overfitting to redundant head data. In contrast, the DS stage leverages the latent representations of scarce images to adaptively synthesize the underrepresented tail data, significantly improving the model's performance on tail-end concepts.

We design comprehensive experiments to validate the effectiveness of ADR in mitigating the LT issue using diverse benchmarks, hallucination metrics, and evaluations with GPT-4. The results demonstrate that ADR significantly improves performance over baseline data and methods, highlighting its strong generalization capabilities. Across all 10 benchmarks, ADR consistently achieves the best results, improving the average performance of LLaVA 1.5 relatively by 2.62%, without increasing the training data volume or introducing additional training. In a nutshell, our contributions are threefold:

- We conduct an in-depth analysis of the long-tail problem in the training data of LVLMs from four key aspects and analyze why mitigating this issue can lead to positive outcomes.
- Our ADR alleviates the LVLM's LT issue and significantly enhances the overall performance of LVLMs without increasing the data scale or introducing any additional training. ADR is also model-agnostic and data-agnostic, which can be easily transferred to any open-source data.
- Comprehensive evaluation including GPT assessments proves the superior performance gain brought by ADR, which serves as a powerful and universal multimodal data modulator.

## 2 RELATED WORK

### 2.1 LARGE VISION-LANGUAGE MODELS

Recently, Large Vision-Language Models (LVLMs) have attracted significant attention. Besides the powerful business models such as GPT-4V, GPT-4o, Gemini, and Claude (OpenAI, 2023; 2024; Team et al., 2023; Anthropic, 2024), many open-source LVLMs exist. Although LVLMs with an integrated overall structure (Wu et al., 2023; Zhan et al., 2024; Team, 2024) have been emerging and

becoming a new trend gradually, traditional aligning-based LVLMs (Liu et al., 2023a; Bai et al., 2023; Zhu et al., 2023; Dai et al., 2024) still make up the majority. With the aid of strong large language models such as LLaMA (Touvron et al., 2023) or Vicuna (Chiang et al., 2023), and a powerful vision encoder such as CLIP (Radford et al., 2021), they managed to align visual comprehension with remarkable language generation capabilities, formulating powerful vision understanding models. However, all of the aforementioned LVLMs still face significant LT issues regardless of their structure or training phases. Therefore, this paper focuses on addressing the LT problems to facilitate the practical application of LVLMs.

## 2.2 DATA DEVELOPMENT OF LVLMS

The instruction-tuning data for VLMs typically includes carefully crafted instructions designed to enhance the general instruction-following capabilities or improve downstream task performance of LVLMs. These instructions are often generated by large language models (LLMs) like GPT-4 (Liu et al., 2024a) or LVLMs such as GPT-4V (Chen et al., 2024b; Yan et al., 2024; Tang et al., 2024). Notably, ShareGPT4V (Chen et al., 2023b) was initially developed from 100K high-quality captions collected from GPT-4V, which were later expanded to 1.2M using a captioning model. Additionally, various data augmentation techniques are employed during LVLM development, such as random cropping and flipping for vision encoders (Ye et al., 2024) and projectors (Li et al., 2023c; Ye et al., 2023a), as well as word- and sentence-level augmentation for instruction tuning (Chen et al., 2024a). However, these augmentation methods often overlook the inherent distribution of the training data, leading to an inability to balance the data distribution effectively.

In addition to data acquisition and augmentation, there is also significant research on data filtering. Several papers have addressed the data diet problem for instruction-tuning in Vision-Language Models (VLMs) (Wei et al., 2023; Liu et al., 2024b; Chen et al., 2024d; Lee et al., 2024). For instance, Paul et al. (2021) introduces two popular importance scores for effective data pruning. Self-Filter (Chen et al., 2024d) adapts gradient as an importance score to train a scoring network, which then filters out important instances. TIVE (Liu et al., 2024b) leverages gradient-based importance scores to design a data filtering pipeline, achieving comparable performance to the original LVLM version while using less data. COINCIDE (Lee et al., 2024) examines the distributional differences between training sets and benchmark data, using a smaller model as a reference to select visual instruction tuning data for more efficient fine-tuning of the target LVLM. However, while existing data filtering and data balancing methods have shown potential in achieving performance on par with the baseline and improving data efficiency, the overall improvements remain modest.

## 2.3 LONG TAIL ANALYSIS OF VLMS

Quantities of work have been done to analyze the long tail problem of traditional VLMs. Some recent studies (Shao et al., 2023; Wang et al., 2022; Zhu et al., 2024) seek to mitigate imbalanced predictions of VLMs by training on additional data from downstream tasks. MetaCLIP (Xu et al., 2023) analyzes the long tail problem of CLIP pre-training data and uses sub-string matching and inverted indexing to balance the pre-training dataset. REAL (Parashar et al., 2024) analyzes the long tail problem of popular image recognition datasets and designs a tail concept replacement method during the inference stage, significantly improving the recognition accuracy of VLMs. However, the LT problem within generative LVLMs is still under-explored.

## 3 ANALYSIS

### 3.1 PRELIMINARY

In the process of LVLM training, the training data typically consists of text-image pairs. However, most of the current visual instruction tuning methods (Liu et al., 2023a; 2024a; Chen et al., 2023b) contain more than one stage, and the format of the training data varies among different stages. Popular methods often consist of 2 stages, i.e. pre-training (alignment) and instruction-tuning (fine-tuning). In the instruction-tuning process of LVLMs, the training data is typically represented as $D = (I, C)$, where $I$ denotes the image input and $C$ represents the corresponding conversation. This paper focuses primarily on the instruction-tuning process of LVLMs, and thus, the **data instances** referenced

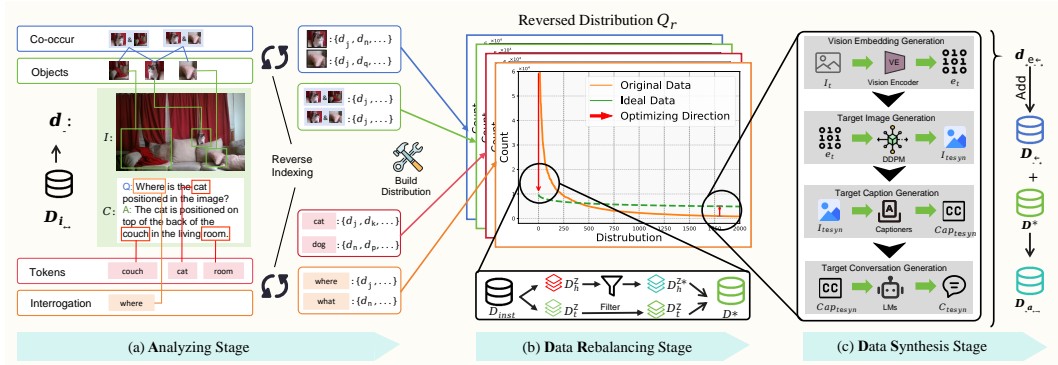

Figure 2: An overview of our **ADR** approach. In the **Analyzing** Stage, we first extract entities—such as tokens, objects, co-occurrences, and interrogations—from the training instances, then construct a quantity distribution using a reverse-indexed entity-to-instance mapping. In the **DR** Stage, we adaptively rebalance the redundant data based on the entity distribution identified in the Analyzing stage. Finally, in the **DS** Stage, we utilize DDPM and the latent representations of scarce image instances to synthesize the underrepresented data.

throughout correspond to $(I, C)$. Each data instance contains **entities** that capture fine-grained semantic information. We extract these entities from different perspectives for further analysis. The complete pipeline of our analysis and approach is illustrated in Figure 2.

## 3.2 ENTITY DISTRIBUTION CONSTRUCTION

To conduct a detailed analysis of the LT problems within LVLMs, we refine our analysis perspective into four aspects, namely Token, Object, Co-occurrence, and Interrogation. Specifically, we conduct the whole analysis procedure by constructing the frequency distribution of entities $Q_e$ from these 4 different perspectives among the whole training set.

**Token** entities are a set of meaningful nouns that are extracted from the text within data instances. $e_t = \{n | n \text{ is a meaningful noun and appears in } C \text{ for } (I, C) \text{ in } D\}$. Technically, we employ a POS parser, stanza (Qi et al., 2020) to extract all nouns from each data instance within the training set, identifying them as token entities. Subsequently, we compute the frequency distribution of all extracted token entities across the entire training set.

**Object** entities represent the objects that truly exist in the image within data instances. $e_o = \{o | o \text{ exists in } I \text{ for } (I, C) \text{ in } D\}$. We initially employ language models (LM), i.e., Llama3 (Touvron et al., 2023) to extract all potential objects from the textual records of each data instance within the training set. The full prompts used to extract object information are detailed in the Appendix C.1. Subsequently, we input the image along with all token entities and LM-extracted objects into a visual grounding model, i.e., GroundingDINO (Liu et al., 2023b) to identify visual objects for each data instance, termed as object entities. Finally, we compute the frequency distribution of all object entities across the entire training set.

**Co-occurrence** entities represent two objects that appear in the same image of one data instance. $e_c = \{(o_1, o_2) | o_1, o_2 \text{ both exist in } I \text{ for } (I, C) \text{ in } D\}$. We utilize the extracted object entities to construct a co-occurrence graph $G(V, E)$. The vertex set V comprises $\{o | o \text{ represents one object entity}\}$, and the edge set $E = \{(o_1, o_2, n) | o_1 \text{ and } o_2 \text{ appear together in the same image for } n \text{ times}\}$. Subsequently, we employ the edge set E as co-occurrence entities to compile the frequency distribution of all co-occurrence entities across the entire training set.

**Interrogation** entities are the questioning methods used in the text within data instances. $e_w = \{q | q \text{ is the questioning method in } C \text{ for } (I, C) \text{ in } D\}$. We employ language models (LM) to extract all methods of posing questions from the data instances, defining them as interrogation entities. We then calculate the frequency distribution of all interrogation entities across the entire training set. We extract all four kinds of entities from LLaVA (Liu et al., 2023a)'s instruction-tuning dataset and the **Top-20** frequently shown entities can be found in the Appendix B.1.

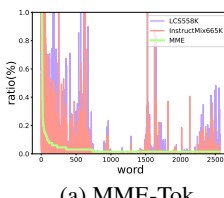 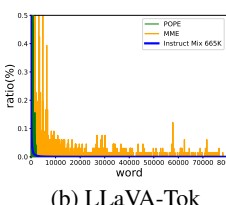 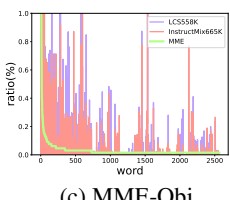 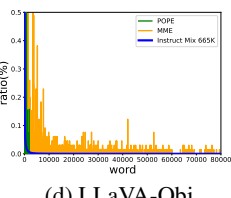

(a) MME-Tok          (b) LLaVA-Tok          (c) MME-Obj          (d) LLaVA-Obj

Figure 3: Long-tail distribution in instruction-tuning and benchmark datasets: (a) Token-level word distribution in MME (Fu et al., 2023). (b) Token-level word distribution in InstructMix665K (Liu et al., 2024a). (c) Object-level word distribution in MME Fu et al. (2023). (d) Object-level word distribution in InstructMix665K (Liu et al., 2024a).

## 3.3 REVERSE INDEXING

To observe the severity of long-tail issues within the training data and build a connection between entities and data instances, we build a reverse indexing dictionary mapping from the entities in four different perspectives backward to the data instances. Subsequently, we use the number of data instances corresponding to each entity as frequency to build the **reversed distribution** $Q_r$ of four perspectives.

Taking LLaVA 1.5 (Liu et al., 2023a)'s instruction tuning data as an example, we count the number of data instance matches for each entity and build a reversed distribution based on the

Table 1: Relative data volume of tail data after reverse indexing. "Tok," "Obj," "Co," and "Int" represent Token, Object, Co-occurrence, and Interrogation, respectively. **%E** denotes the percentage of tail entities, while **%DI** indicates the percentage of tail data instances.

| Data | Level | thres | % E | % DI |
|------|-------|-------|------|------|
| **LLaVA** | Tok | 120 | 98.7 | 10.0 |
| | Obj | 304 | 98.0 | 10.0 |
| | Co | 24 | 92.7 | 25.0 |
| | Int | 4895 | 99.6 | 10.0 |
| | Avg. | - | **97.25** | **13.75** |

mapping data. The tail data's thresholds and relative data volume are shown in Table 1. Surprisingly, among 4 perspectives, an average of **97.25%** entries account for only **13.75%** data instances on average, which can partially illustrate the scarcity of tail data and severity of the long-tail problem existing in LVLMs' training data.

## 3.4 WHY MITIGATING THE LT PROBLEM CAN BOOST THE PERFORMANCE OF LVLMS?

- **Unbalanced training data**. We begin by illustrating the long-tail (LT) problem in training data through a detailed analysis. Using LLaVA 1.5's instruction-tuning data as an example (Liu et al., 2023a), the distribution curve shown in Figure 3 reveals a pronounced imbalance in the distribution of entities. The head (high-frequency) entities appear significantly more often than those in the tail (low-frequency). Although real-world data distributions are typically long-tailed, balanced data is more effective in facilitating the learning of tail concepts (Parashar et al., 2024). The unbalanced entity distribution in training data limits models' ability to adequately learn these tail concepts. Furthermore, several studies (Liu et al., 2024b; 2023a) have shown that current LVLM instruction-tuning datasets contain substantial redundancy, indicating that reducing part of the training data has minimal impact on performance and helps mitigate overfitting. Our experimental results further support this finding.

- **Tail data accounts for more failed cases**. Moreover, we found that tail data accounts for more failed cases, indicating that it is also necessary to supplement the scarce tail data. We evaluate the model's performance on two popular benchmarks, POPE and MME, and analyze the failed cases, as incorrect responses generated by LVLMs often reveal the model's weaknesses. Consequently, we rank the failed cases based on the entity distribution of LLaVA's instruction-tuning data and extract the bottom (tail) **50%** of these cases. Next, we measure the number of entities and data instances corresponding to these errors. The results, presented in Table 2, reveal that the tail 50% of failed cases cover over **99%** of entities and account for an average of **68.65%** of training instances. Additionally, we plot a cumulative error curve with entity distribution on the horizontal axis, ordered from most to least frequent. The cumulative curve is shown in Figure 4. Further, we analyze the distribution locations of incorrect and correct answers generated by LLaVA 1.5 (Liu et al., 2023a) in POPE (Li et al., 2023d) and MME (Fu et al., 2023). The results are shown in the

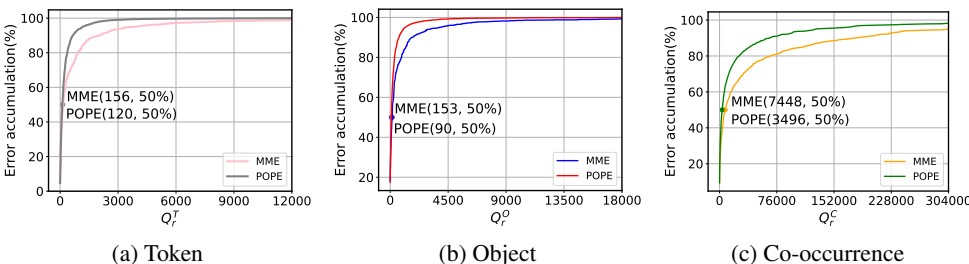

|        (a) Token        |        (b) Object        |        (c) Co-occurrence        |

Figure 4: Error accumulate curve of popular benchmarks based on the training data distribution. (a) Token-level word distribution in MME (Fu et al., 2023) and POPE (Li et al., 2023d). (b) Object-level word distribution in MME and POPE. (c) Co-occurrence-level word distribution in MME and POPE.

Appendix B.3. Interestingly, despite the influence of the averaging effect, the average locations of wrong answers tend to be positioned further towards the tail of the distribution compared to correct answers, indicating the promoting space of tail instances.

- **Distribution varies between train and test data.** Besides, the distribution of train and test data is also different. In statistics-based deep learning, it is assumed that the training data maintains the same distribution as the evaluation data. Based on the intuition that a larger bias between distributions can result in performance loss, we examine the differences between the entity distributions of the training and evaluation data. We select LLaVA 1.5's training data (Liu et al., 2023a), as well as the evaluation data from POPE (Li et al., 2023d) and MME (Fu et al., 2023), and plot the distribution curves for both the training and evaluation data on the same scale in a single graph. The resulting co-distribution is presented in Figure 3. Notably, a clear difference between the distributions of the evaluation and training data can be observed.

## 4 APPROACH

### 4.1 DATA REBALANCING STAGE

As shown in Figure 3. Entities among the four perspectives suffer from LT problems, where the head part of the distribution turns out to be exponential. To mitigate the LT problem, our ADR starts by alleviating the redundancy problem existing in training data. Concretely, we do this by flattening the exponential distribution and decimating the duplicated entities.

Table 2: Relative training data volume for the tail **50%** of failed cases. "Tok," "Obj," and "Co" refer to Token, Object, and Co-occurrence, respectively. **%E** denotes the percentage of tail entities, while **%DI** represents the percentage of tail data instances.

| **Data** | **Level** | **thres** | **% E** | **% DI** |
|---|---|---|---|---|
| **MME** | Tok | 156 | 99.98 | 75.23 |
|  | Obj | 153 | 99.83 | 51.33 |
|  | Co | 7448 | 99.02 | 69.62 |
| **POPE** | Tok | 120 | 99.98 | 80.50 |
|  | Obj | 90 | 99.71 | 59.76 |
|  | Co | 3496 | 99.54 | 75.45 |
| Avg. | - | - | 99.68 | 68.65 |

#### 4.1.1 PROBABILITY DICTIONARY CONSTRUCTION

We conduct the DR stage by initially settling down the resampling ratios of redundant data instances. First, we use the entity distribution construction method mentioned in Section 3.2 to construct the distribution dictionary $Q_e$ for each entity within all selected perspectives $C$. Subsequently, we construct the reverse indexing dictionary and use the data instance numbers mapped by entities $N_e$ as frequency to build a reversed distribution $Q_r$, which is used to distinguish between head and tail data and calculate the sampling ratio $p_s$. A threshold $\tau$ distinguishes the head and tail data. $\tau$ is an entity's position in $Q_r$ while entities before $\tau$ count for a small ratio of all entities but are mapped to massive data instances. We set $\tau$ as indicated in Table 1, consistent with the values used in the analysis stage. For an entity $e$ of perspective $x$, we set the probability of sampling by $P_{e_x} = \tau_x / N_{e_x}$.

After constructing the probability dictionary of each entity $e$, we start from $Q_e$ to sample the selected data. Since each data instance contains several entities in $Q_e$, we sample each entity among one instance. So we introduce a new hyperparameter $n_p$, which means the data instance with the total

---

**Algorithm 1** Pseudo Code for **D**ata **R**esampling

---

```
1  D_bal=[]              # D_bal is the rebalanced data, a.k.a. D*
2  for pers in C:        # build prob dict
3      entity_dist = entity_distribution_construction(D,pers)
4      prob_dict[pers] = {ent:tau[pers]/entry_dist[ent] for ent in
           entry_dict.keys()} # tau is threshold for different entities
5  for instance in D: # do distribution balancing
6      pass_cnt = 0
7      for pers in C: # C is selected perspectives
8          for entity in instance['entities'][pers]:
9              if random.random() < prob_dict[pers][entity]:
10                 pass_cnt += 1
11                 break
12     if pass_cnt > n_p and random.random() < alpha:
13         D_bal.append(instance) # n_p, alpha: hyperparameters
```

---

number of sampled entities over $n_p$ is selected. We conduct this procedure over the full dataset, and the final selected core set is denoted as $D^*$. The detailed method is demonstrated in Algorithm 1.

## 4.2 DATA SYNTHESIS STAGE

Despite the presence of redundancy in the head entities, the issue of scarcity in the tail entities still persists. To alleviate the issue of scarcity, we design the data synthesis methods from the perspective of vision and language. Figure 2 displays the full data adjusting framework.

### 4.2.1 LANGUAGE DATA SYNTHESIS

We introduce a method for synthesizing the tail data at the language level. The core idea is to replace head concepts with tail ones. First, we use WordNet (Fellbaum, 1998) to extract all synonyms of token entities and construct a mapping system. For each head instance, we extract its linguistic entities, search for their synonyms in the mapper, and filter out the head ones. Next, we feed the original head conversation into a language model (LM) and prompt the LM to rewrite the conversation using the selected tail synonyms. Full prompts to instruct LMs can be found in Appendix C.2. It's important to note that certain stop words, such as "image" should not be replaced.

### 4.2.2 DIFFUSION BASED VISUAL DATA SYNTHESIS

In addition to the language synthesis method, we propose a more comprehensive approach that allows tail data synthesis from multiple perspectives. The scarcity of objects and co-occurring objects can be addressed by editing tail images into different styles without altering the key entities. Meanwhile, a rewriting process can resolve the scarcity of tokens and interrogation methods. In our analysis of the long-tail problem, we determine whether a data instance is selected using $P_e$ and $n_p$.

However, in certain cases, the probability $P_e$ may exceed 1. The absolute value of $P_e$ also provides some insight into the scarcity of a data instance. This value can be used to decide how many instances to synthesize. Notably, **76%** of the synthetic data has a $P_e$ value of **less than 5**, so we utilize this method to determine the synthetic quantity for each data instance. The synthetic quantity for each data instance $d = (I, C)$ can be calculated as:

$$P_d^* = \max_{e,x} P_{e_x}; N_{d,aug} = \begin{cases} 0 & \text{if } P_d^* < 1, \\ \lfloor \sqrt{P_d^*} \rfloor & \text{if } 1 \le P_d^* < 5, \\ 2 & \text{if } 5 \le P_d^*, \end{cases} \quad (1)$$

After setting down the synthesis quantity of each tail instance, we utilize diffusion models (Croitoru et al., 2023; Ho et al., 2020; Rombach et al., 2022) to do visual generation for tail instances. Given a tail instance $d_t = (I_t, C_t)$, our objective is to generate an image similar to $I$ and produce corresponding instruction data, specifically in the form of conversations. To achieve this, we utilize Stable-Diffusion-V2 as our DDPM model, which can generate a new image $G(g(t), n)$ based on

Table 3: **Comparison with models trained with different methods on different benchmarks.** IT represents the number of training instances used during instruction tuning. +DR denotes the results after the data rebalancing stage, while +DS represents the results following the data synthesis stage. Benchmark names are abbreviated due to space limits. *: ShareGPT4V's instruction tuning stage refers to the 2nd stage (3 in total). The best results are indicated in **bold**.

| Method | IT* | VQA$^{v2}$ | SQA$^I$ | MMMU | MME$^P$ | MMS | VQA$^T$ | GQA | QB$^2$ | VQA$^{OK}$ | MMB |
|--------|-----|-----------|---------|------|---------|-----|---------|-----|--------|-----------|-----|
| LLaVA 1.5 | 665.0K | 76.6 | 69.3 | 35.3 | 1510.7 | 33.5 | 46.0 | 61.9 | 47.3 | 53.2 | 64.3 |
| +DR | 581.0K | 76.9 | 69.5 | 34.8 | 1470.6 | 33.8 | 46.0 | 62.8 | 46.8 | **55.3** | **65.5** |
| +DR +DS | 665.0K | **76.9** | **70.2** | **36.3** | **1511.3** | **35.5** | **46.5** | **62.9** | **49.6** | 55.1 | 65.0 |
| ShareGPT4V | 1246.0K | 78.6 | 68.9 | 35.1 | 1560.4 | 34.7 | 50.2 | 63.3 | 44.2 | 54.0 | 68.0 |
| +DR | 1168.0K | 78.7 | 68.6 | 35.7 | 1542.3 | 35.0 | **50.9** | **63.9** | 44.9 | 56.7 | 67.9 |
| +DR +DS | 1246.0K | **78.7** | **69.4** | **36.1** | **1564.9** | **35.5** | 50.9 | 63.7 | **45.7** | **57.9** | **68.8** |

natural language descriptions t, where $n \sim N(0, 1)$ represents the sampling noise. However, since it's required to generate another style for $I_t$ without interference from text inputs in DDPM models, similarly to Feng et al. (2023), we adopt CLIP embeddings $e_t = f_{CLIP}(I_t)$ to replace $g(t)$. Thus, the synthetic image can be generated as:

$$I_{t,syn} = G(f_{CLIP}(I_t), n) \tag{2}$$

After generating the synthesis image $I_{t,syn}$, the next step is to obtain a descriptive conversation paired with the image to serve as the instruction tuning data instance. For this purpose, we utilize an off-the-shelf vision captioner to generate captions $Cap_{t,syn}$ for $I_{t,syn}$. Subsequently, we extend the captions into conversations, as required by visual instruction tuning. During this stage, we use a language model (LM) to expand the captions into full conversations, with the prompts for expansion provided in Appendix C.2. This process enables us to effectively synthesize scarce data instances, helping to alleviate the LT problem from all four perspectives.

## 5 EXPERIMENTS

### 5.1 BASELINE MODELS

In this paper, we select LLaVA1.5 and ShareGPT4V as our baseline method. We conduct our LT mitigating method (ADR) on these two models to verify the effectiveness.

**LLaVA 1.5** (Liu et al., 2024a; 2023a) represents a novel end-to-end trained large multimodal model that combines a vision encoder and Vicuna for general-purpose visual and language understanding, achieving impressive chat capabilities.

**ShareGPT4V** (Chen et al., 2023b) uses the adjusted training data obtained by GPT-4 and post-trained ShareCapioner and improves the performance of existing VLMs. Though they focus on data adjustment either, the long-tail problem remains, that is, our method is orthogonal to ShareGPT4V.

### 5.2 BENCHMARKS

To evaluate changes in the model's overall capabilities and performance, we utilized a comprehensive set of widely recognized benchmarks, spanning a broad range of academic Visual Question-answering (VQA) tasks and recent benchmarks designed to test the extensive abilities of LVLMs. The VQA series benchmarks (Goyal et al., 2017; Marino et al., 2019; Singh et al., 2019) represent traditional, comprehensive VQA tasks. GQA (Hudson & Manning, 2019) evaluates multiple reasoning skills and spatial understanding, which presents a greater challenge. The MME Benchmark (Fu et al., 2023) assesses the comprehensive capabilities of LVLMs through a series of carefully crafted questions spanning 14 distinct sub-tasks. MMBench and MMBench-CN (Liu et al., 2023c) are designed to assess the model's vision-based reasoning and perceptual abilities in both English and Chinese. MMSTAR (Chen et al., 2024c) and SEED (Li et al., 2023b) evaluate the model's comprehensive ability from different aspects. ScienceQA (Lu et al., 2022) evaluates LVLMs on multimodal, multiple-choice science questions, while MMMU (Yue et al., 2023) tests LVLMs across multiple disciplines, requiring college-level subject knowledge and sophisticated reasoning. Finally, Q-Bench (Wu et al.,

Table 5: Performance comparison across existing data balancing methods. The best results are indicated in **bold**, and the second-best results are underlined.

| Method | IT | VQA$^{v2}$ | GQA | VizWiz | SQA$^I$ | VQA$^T$ | POPE | MME | MMB | MMB$^{CN}$ | MMS | QB$^2$ |
|---|---|---|---|---|---|---|---|---|---|---|---|---|
| LLaVA 1.5 | 665.0K | 76.6 | 62.0 | 50.0 | 66.8 | 46.0 | 85.9 | 1510.7 | 64.3 | 55.3 | 33.5 | 47.3 |
| Random | 581.0K | 76.9 | 62.3 | 55.6 | 68.8 | 46.4 | 87.2 | 1472.4 | 65.5 | 55.7 | 34.9 | 47.3 |
| perplexity | 581.0K | 76.7 | 62.3 | 55.3 | 68.8 | 45.8 | 86.8 | 1484.4 | 63.7 | 54.7 | 33.7 | 47.1 |
| COINCIDE | 133K | 76.5 | 59.8 | 46.8 | 69.2 | - | 86.1 | 1495.6 | 63.1 | 54.5 | - | - |
| Our-Balance | 581.0K | 76.9 | 62.8 | 53.8 | 69.5 | 46.0 | 87.2 | 1470.6 | 65.5 | 55.7 | 33.8 | 46.8 |
| Ours | 665.0K | 76.9 | 62.9 | 56.8 | 70.2 | 46.5 | 86.9 | 1511.3 | 65.0 | 56.0 | 35.5 | 49.6 |

2024) focuses on assessing low-level perception. All benchmarks we used and their abbreviations can be found in Appendix A.1.

### 5.3 RESULTS FOR COMPREHENSIVE EVALUATION

The results on the 10 selected benchmarks are shown in Table 3. By balancing the training data while retaining 87% of its original scale, our method outperforms the baseline on most benchmarks. After the DS stage, with the same data scale as LLaVA 1.5, our method achieves an average relative improvement of 2.26%. Notably, it outperforms the baseline by an average of **5.42%** relatively on challenging benchmarks like MMStar and Qbench-2.

Also, we displayed the comparison between our method and popular data-balancing methods such as random, perplexity (Marion et al., 2023), and COINCIDE (Lee et al., 2024) in ta-

Table 4: Tail concept prediction accuracy (%) on ScienceQA-IMG (Lu et al., 2022) dataset. Tail@$k$% (simplified as @$k$), head@$k$% (simplified as H@$k$), and overall accuracy are reported. +DR denotes the results after data rebalancing, while +DS represents the results following the data synthesis stage. **Bold** numbers represent the best results across all methods.

| Methods | FT | ScienceQA | | | | | |
|---|---|---|---|---|---|---|---|
| | | @5 | @10 | @15 | @20 | H@80 | Overall |
| LLaVA 1.5 | 665.0K | 67.9 | 70.0 | 67.9 | 68.5 | 74.6 | 69.3 |
| +DR | 581.0K | 69.2 | 69.7 | 67.8 | 68.5 | 76.2 | 69.5 |
| +DR +DS | 665.0K | **70.1** | **70.5** | **68.3** | **69.0** | **78.6** | **70.2** |

ble 5. However, COINCIDE hasn't made their code yet, so we obtained the result from their papers. Our method consistently outperforms the baselines across the majority of benchmarks, which cover a wide range of comprehensive tasks. Additionally, our approach focuses on mitigating LT issues and is orthogonal to most existing data balancing and augmentation methods. This means it can be applied alongside those techniques to achieve even better performance across various benchmarks.

### 5.4 PERFORMANCE ON TAIL INSTANCES

In addition to the main results on comprehensive benchmarks, we assess the model's performance on tail concepts to validate the effectiveness of improving tail concept performance. We selected ScienceQA (Lu et al., 2022) and VQAV2 (Goyal et al., 2017) to evaluate performance on tail data. First, we applied the same method described in Section 3.2 to extract entities from the selected benchmark data and build their reverse indexed distribution. Subsequently, for each data instance, we calculated the average distribution position across each perspective and determined whether the data instance falls into the tail category by $\mathbb{1}(average(L_i) > \tau_R)$. Here, $\mathbb{1}$ represents an indicator function. We set different thresholds to make sure to get different ratios of tail data. We then extract the tail data instances of different rations and evaluate the performance accordingly. The complete results for tail performance are presented in Table 4 and Figure 1(b). As shown in the table, our method effectively improves tail performance **without compromising the head or overall performance**, demonstrating the efficacy of our approach.

## 6 ABLATION STUDY

### 6.1 ABLATION OF DIFFERENT COMBINATIONS OF PERSPECTIVES

To determine the most effective rebalancing and synthesis method, we train the model using data processed with different combinations of perspectives and subsequently evaluate the target model. The results are presented in Figure 5 (a). We assess these combinations based on average perfor-

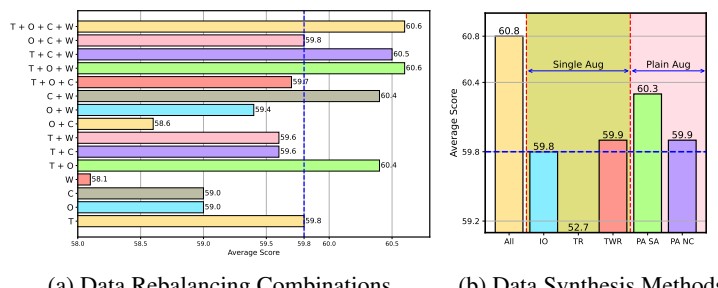

(a) Data Rebalancing Combinations      (b) Data Synthesis Methods

Figure 5: Ablation study on data rebalancing combinations (a) and synthesis methods (b). T, O, C, and W refer to Token, Object, Co-occurrence, and Interrogation respectively. The values displayed in the graph represent average scores across a variety of comprehensive benchmarks. The blue dashed line indicates the baseline performance of LLaVA 1.5.

mance across different benchmarks, the number of top-ranked results, and performance stability. While several combinations achieved the highest average performance, the full combination of all perspectives proved to be the most stable, as it ranked first in both the number of top performances and stability. Detailed results can be found in the Appendix A.2.

## 6.2 ABLATION OF SYNTHESIS METHODS

In addition to the perspective combination selection during the Data Rebalancing stage, we present our ablation study on different synthesis methods. Synthesis is performed on the same rebalanced data checkpoint across all perspectives to determine which method is the most effective.

We select six synthesis or augmenting methods, divided into language synthesis methods (Sec.4.2.1) and vision synthesis methods (Sec.4.2.2). The following methods are tested: **All**: Tail instances are selected from all four perspectives, with full visual data synthesis (Sec.4.2.2) applied. **Image Only (IO)**: Tail instances are selected from all four perspectives, applying visual data synthesis (Sec.4.2.2), but the conversation text remains unchanged. **Token Rewrite (TR)**: Full language data synthesis (Sec.4.2.1) methods are applied. **TW Rewrite (TWR)**: Tail instances are selected based on Token and Interrogation perspectives, and the conversations are rewritten using a language model (LM). **PlainAug SimpAdd (PA SA)**: Tail data are selected from all four perspectives, and simple resampling is applied. **PainAug NewCap (PA NC)**: Tail data are selected from all four perspectives, followed by re-captioning, with the new captions incorporated into conversations using the same method with Sec.4.2.2.

We use these synthesis methods to restore the data to 665K and test which checkpoint yielded the best performance using the same volume of training data. We assess these methods based on average performance across different comprehensive benchmarks. The results are displayed in Figure 5(b), while detailed results can be found in the Appendix A.2.

## 7 CONCLUSION

In this paper, we analyze the relation between the LT problem existing in LVLM instruction tuning data from multiple perspectives and make the first attempt to mitigate it. Our analysis reveals that unbalanced data can result in a performance gap between the head data and the tail data and therefore harm the model's performance. Based on them, we develop an Adaptive Data Refinement Framework (ADR), which first balances the redundant head data from different perspectives and then adjusts the unbalanced tail data volume with a specific focus on tail distribution. Experimental results demonstrate that our method improves the tail performance and overall performance without harming the head part performance, resulting in a more robust LVLM instruction-tuning pipeline.

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

## A   DETAILES OF EVALUATION METHODS

### A.1   BENCHMARKS

All benchmarks we used and their abbreviations are introduced as follows.

$VQA^{v2}$: VQA-V2 (Goyal et al., 2017); $VQA^T$: TextVQA (Singh et al., 2019); $VQA^{OK}$: OK-VQA (Marino et al., 2019); REF: Ref-COCO (Kazemzadeh et al., 2014); REF+: Ref-COCO+ (Kazemzadeh et al., 2014); GQA (Hudson & Manning, 2019); $SQA^I$: ScienceQA-IMG (Lu et al., 2022); FLIK: flickr30k (Young et al., 2014); POPE (Li et al., 2023d); SEED: SEED Bench (Li et al., 2023b); $SEED^{v2}$: SEED Bench v2 (Li et al., 2023a); MMMU (Yue et al., 2023); $MME^P$ (Fu et al., 2023); $MMB^{CN}$: MM-Bench Chinese (Liu et al., 2023c); MMB: MM-Bench English (Liu et al., 2023c); MMS (Chen et al., 2024c); $QB^2$: Q-Bench 2  (Wu et al., 2024).

### A.2   SUPPLYMENTARY RESULTS OF ABLATION STUDY

We conducted an ablation study on different balancing combinations and synthesis methods. In the ablation study of different rebalancing combinations, we conduct the DR stage using different combinations of four perspectives, i.e., one or more from (Token, Object, Co-occurrence, and Interrogation) to validate the effectiveness of different perspectives. The detailed results of the balancing ablation experiment are presented in Table 6. We found that combining all perspectives yields the best performance in terms of both the number of top results and performance stability.

Additionally, we conducted an ablation study on different synthesis methods. The results of the augmentation and synthesis experiments are presented in Table 7. It is clear that synthesizing from **ALL** perspectives (as outlined in Section 4.2.2) yields the best performance.

## B   DETAILS OF ANALYZING STAGE

### B.1   EXAMPLES OF ENTITIES

Entities are extracted from four perspectives: Token, Object, Co-occurrence, and Interrogations. The top 20 entities from LLaVA 1.5's instruction-tuning data are displayed in Figure 6.

### B.2   ENTITY DISTRIBUTION CONSTRUCTION

In this work, we construct the entity distribution using both the pre-training and instruction-tuning data from LLaVA 1.5, specifically LCS558K and Instructmix665K. To further compare the differences between the training and test data, we also incorporate part of the distribution from POPE and MME in the same figure. The complete results are shown in Figure 7.

### B.3   ANALYSIS OF FAILED CASES

We experiment to observe the distribution location of failed cases. We first extract all entities within the failed cases and calculate the max, min, and average location of these entities in the pertaining distribution. Also, we calculate the distribution locations of the correct cases as well to compare. The results are shown in Table 8. As shown in the table, it is easy to discover that the failed cases are positioned further behind the correct ones in the distribution.

## C   PROMPTS

### C.1   OBJECT INFORMATION EXTRACTION

In this section, we release all of our prompts for guiding language models (LM) to do specific tasks. Firstly during the analyzing stage, we utilize the LM to extract object information from the text within data instances at the very first step during object entity extraction. This part of the prompt we used to guide LM is illustrated in Figure 8.

Table 6: Full results of ablation study on different combinations of perspectives. T, O, C, and W refer to Token, Object, Co-occurrence, and Interrogation respectively. The best results are indicated in **bold**, and the second-best results are underlined.

| T | O | C | W | IT | VQA$^{v2}$ | VQA$^T$ | VQA$^{OK}$ | GQA | SQA | SQA$^I$ | REF | REF+ | FLIK | POPE | SEED | Avg. |
|---|---|---|---|---|---|---|---|---|---|---|---|---|---|---|---|---|
| | baseline | | | 665.0K | 76.6 | 46.0 | 53.2 | 61.9 | 70.4 | 69.3 | 29.4 | 28.5 | 74.9 | 86.9 | 60.6 | 59.8 |
| ✓ | | | | 488.1K | 76.5 | 46.6 | 55.3 | 62.3 | 70.8 | 69.2 | 28.5 | 28.1 | 73.8 | 86.7 | 60.2 | 59.8 |
| | ✓ | | | 197.9K | 74.6 | 44.0 | 50.4 | 61.3 | 69.9 | 67.9 | 30.8 | 29.7 | 74.1 | 86.3 | 59.3 | 59.0 |
| | | ✓ | | 242.4K | 75.2 | 43.3 | 47.3 | 61.3 | 70.0 | 68.5 | 31.4 | 29.8 | 76.2 | 86.8 | 59.0 | 59.0 |
| | | | ✓ | 176.3K | 73.9 | 43.0 | 46.3 | 60.7 | 69.5 | 66.7 | **32.3** | **31.7** | 71.9 | 85.6 | 57.4 | 58.1 |
| ✓ | ✓ | | | 534.2K | 76.7 | 47.1 | 55.6 | 62.8 | 71.4 | 68.1 | 30.3 | 29.1 | 75.4 | 86.9 | 60.9 | 60.4 |
| ✓ | | ✓ | | 553.4K | 75.7 | 44.5 | 52.8 | 62.0 | 70.8 | 68.4 | 30.4 | 29.2 | 75.1 | 86.4 | 59.9 | 59.6 |
| ✓ | | | ✓ | 521.5K | 75.7 | 44.5 | 52.8 | 62.0 | 70.8 | 68.4 | 30.4 | 29.2 | 75.1 | 86.4 | 59.9 | 59.6 |
| | ✓ | ✓ | | 276.9K | 75.4 | 44.6 | 46.8 | 61.7 | 69.0 | 66.4 | 30.6 | 29.4 | 74.2 | 87.1 | 59.3 | 58.6 |
| | ✓ | | ✓ | 318.3K | 75.7 | 44.6 | 50.9 | 61.8 | 71.5 | 69.0 | 29.9 | 29.0 | 74.9 | 86.8 | 59.6 | 59.4 |
| | | ✓ | ✓ | 349.9K | 76.8 | 46.8 | 54.4 | 62.5 | 71.5 | 68.8 | 29.9 | 29.2 | 75.7 | 86.8 | **61.5** | 60.4 |
| | ✓ | ✓ | ✓ | 375.9K | 76.2 | 45.3 | 54.4 | 62.8 | 70.7 | 67.6 | 29.7 | 28.8 | 74.3 | 86.8 | 60.1 | 59.7 |
| ✓ | | ✓ | ✓ | 575.5K | 76.8 | 46.7 | **56.7** | 62.4 | 71.2 | 68.8 | 30.1 | 29.1 | 75.9 | 87.2 | 61.2 | 60.6 |
| ✓ | ✓ | | ✓ | 559.3K | 76.7 | 46.9 | 52.5 | 62.3 | 71.6 | 69.2 | 30.8 | 30.0 | 76.6 | 87.4 | 61.0 | 60.5 |
| ✓ | ✓ | ✓ | | 561.5K | 76.8 | 47.2 | 50.0 | 62.3 | 71.7 | 69.9 | 28.8 | 28.1 | 75.6 | 86.6 | 60.6 | 59.8 |
| ✓ | ✓ | ✓ | ✓ | 581.7K | 76.9 | 46.0 | 55.3 | 62.8 | 71.4 | 69.5 | 30.2 | 29.7 | 76.2 | 87.2 | 61.0 | 60.6 |

Table 7: Full results of ablation study on different augmentation methods. Methods are introduced in Sec.6.2. The best results are indicated in **bold**, and the second-best results are underlined.

| Method | IT | VQA$^{v2}$ | VQA$^T$ | VQA$^{OK}$ | GQA | SQA | SQA$^I$ | REF | REF+ | FLIK | POPE | SEED | Avg. |
|---|---|---|---|---|---|---|---|---|---|---|---|---|---|
| ALL | 665.0K | 76.9 | 46.5 | 55.1 | 62.9 | **72.0** | **70.2** | 30.5 | 29.9 | 76.2 | 86.9 | 61.3 | 60.8 |
| Image Only | 665.0K | 76.9 | 46.5 | **57.2** | 62.5 | 68.8 | 68.4 | 30.6 | 30.2 | 75.9 | 87.3 | 53.8 | 59.8 |
| Token Rewrite | 665.0K | 76.9 | 46.1 | 49.2 | 62.4 | 70.6 | 68.6 | **32.3** | **31.3** | 0.6 | 87.4 | 54.1 | 52.7 |
| TW Rewrite | 665.0K | 76.9 | **46.9** | 54.9 | 62.5 | 68.9 | 68.7 | 31.0 | 30.3 | **77.5** | 87.5 | 53.7 | 59.9 |
| PlainAug SimpAdd | 665.3K | 76.8 | 46.2 | 56.0 | **63.0** | 71.7 | 69.3 | 29.3 | 28.5 | 74.1 | 86.6 | **61.7** | 60.3 |
| PlainAug NewCap | 665.3K | 76.8 | 46.7 | 54.6 | 62.1 | 68.5 | 69.4 | 31.1 | 30.7 | 77.3 | 87.7 | 54.1 | 59.9 |

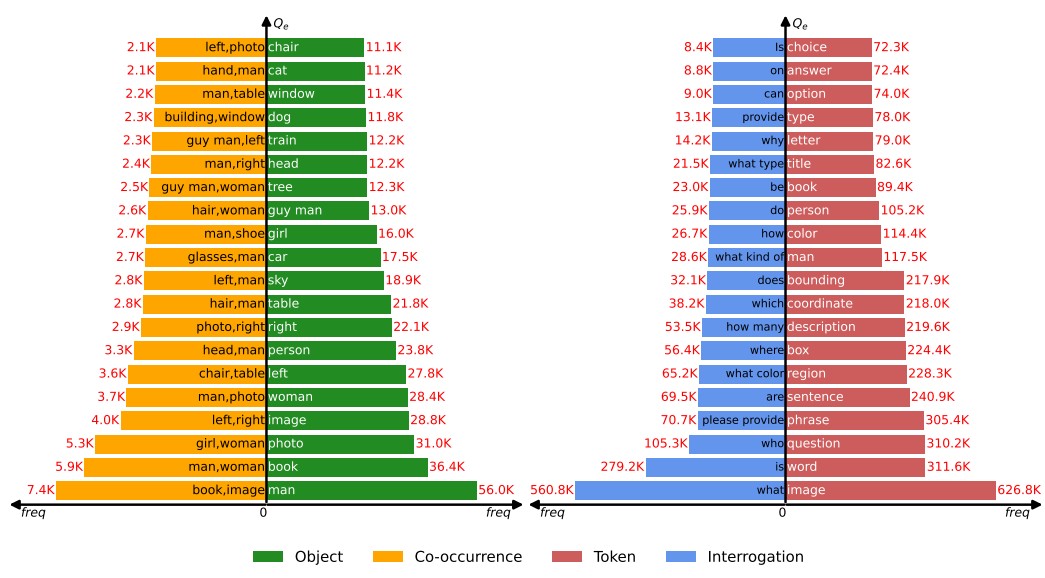

Figure 6: Top 20 entities of LLaVA 1.5's instruction-tuning data.

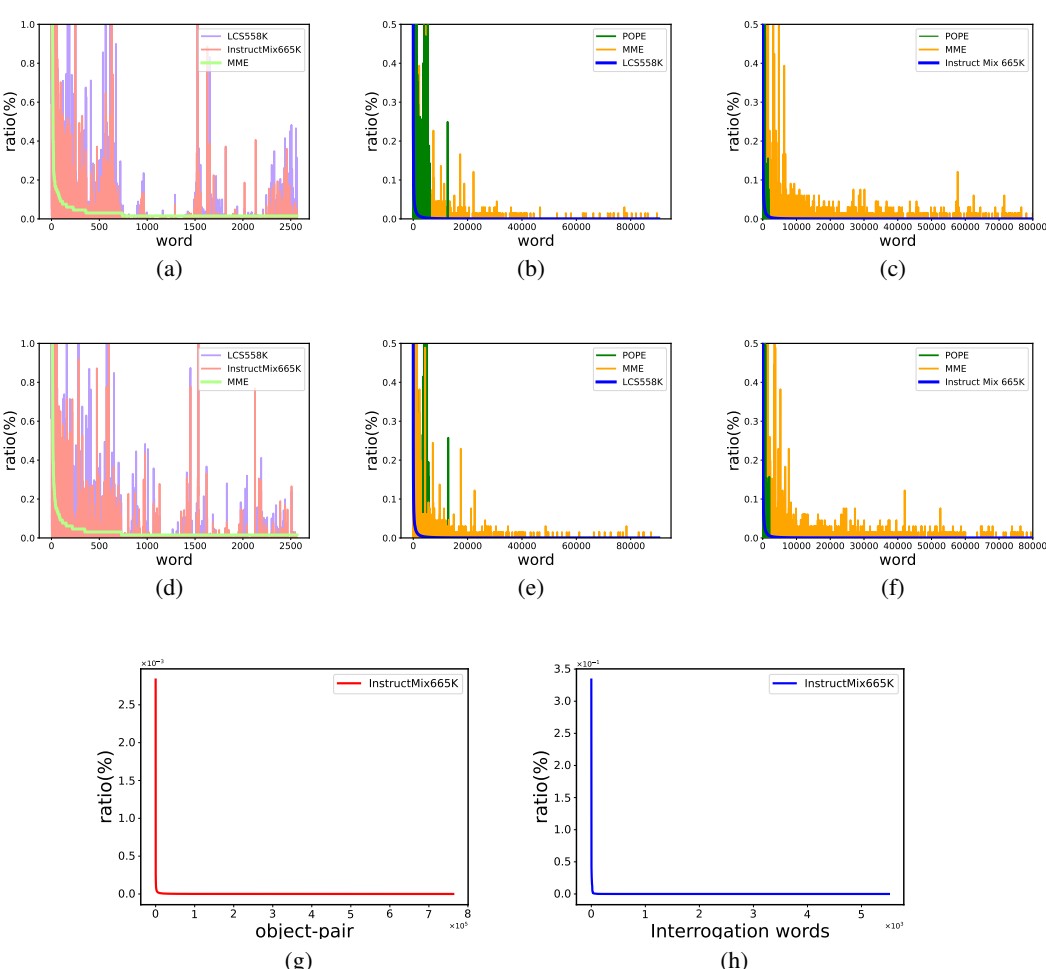

Figure 7: Long-tail distribution in instruction-tuning and benchmark datasets: (a) Token-level word distribution in MME (Fu et al., 2023). (b) Token-level word distribution in LCS558K (Liu et al., 2024a). (c) Token-level word distribution in InstructMix665K (Liu et al., 2024a). (d) Object-level word distribution in MME Fu et al. (2023). (e) Object-level word distribution in LCS558K (Liu et al., 2024a). (f) Object-level word distribution in InstructMix665K.(Liu et al., 2024a). (g) Co-occurrence distribution in InstructMix665K (Liu et al., 2024a). (h) Interrogation distribution in InstructMix665K (Liu et al., 2024a).

Table 8: Distribution locations of entities in correct and incorrect answers for POPE and MME, generated by LLaVA 1.5. "Tok," "Obj," and "Co" refer to Token, Object, and Co-occurrence, respectively, while "W" and "C" represent wrong and correct answers, respectively. The gray rows (■) indicate the relative displacement of incorrect concepts in the distribution compared to correct concepts.

| Methods | MME | | | | | | POPE | | | | | |
|---|---|---|---|---|---|---|---|---|---|---|---|---|
| | Tok-C | Tok-W | Obj-C | Obj-W | Co-C | Co-W | Tok-C | Tok-W | Obj-C | Obj-W | Co-C | Co-W |
| Max | 9738 | 10377 | 2708 | 3222 | 247315 | 257107 | 2242 | 2772 | 1085 | 1100 | 130043 | 141722 |
| | | +639 | | +514 | | +9792 | | +30 | | +15 | | +11679 |
| Min | 1 | 1 | 60 | 131 | 12732 | 20741 | 1 | 1 | 17 | 21 | 926 | 1033 |
| | | +0 | | +71 | | +8009 | | +0 | | +4 | | +107 |
| Mean | 1035 | 1068 | 842 | 1035 | 71123 | 79104 | 313 | 340 | 319 | 336 | 27457 | 30989 |
| | | +33 | | +193 | | +7981 | | +27 | | +17 | | +3532 |

**Instruction:**

You are an AI visual assistant, and you are provided with a single image accompanied by a piece of conversation describing the same image. Your task is to identify and return the objects present in the image based on the conversation. Please follow these guidelines:

1.The output must contain only the base form of the words describing the objects (e.g., return "dog" instead of "dogs" or "doggy").
2.The objects should be listed as comma-separated values (e.g., a,b,c,…).
3.Ensure that the answer consists solely of the object list, with no additional words.
4.The objects can be nouns or noun phrases.

**Fewshots:**

Input 1:
Human: Write a terse but informative summary of the picture.
GPT: A grey watch with an army-style strap.
Expected Answer 1: "watch"
Input 2:
Human: Give a brief description of the image.
GPT: The divine queen in her elaborate masks canvas print featuring the face and hands of a woman with red hair.
Expected Answer 2: "divine queen, mask"
Input 3:
Human: Where is the cat positioned in the image?
GPT: The cat is positioned on top of the back of the couch in the living room.
Human: What is the cat doing in the image?
GPT: The cat is coming out from some curtains onto the couch and is sitting or standing on top of it.
Human: What is the color of the curtains near the cat?
GPT: The color of the curtains near the cat is red.
Human: Besides the cat, what else can be seen on the couch?
GPT: There are lots of pillows on the couch besides the cat.
Human: What kind of room does the couch appear to be in?
GPT: The couch appears to be in a living room setting.
Expected Answer 3: "cat, couch, pillow, curtain, living room"

Figure 8: Complete prompts used to guide the language model in extracting object information.

## C.2 CONVERSATION REWRITE

We leverage language models (LMs), LLaMA3 (Touvron et al., 2023) to rewrite our conversations. During the Data Synthesis (DS) Stage, synthetic data and captions are generated using diffusion models and captioning models. Once the image and its corresponding caption are obtained, we employ the LM to transform the caption into a conversation. The prompt used to guide the LM is shown in Figure 9. The captioning models we used are LLaVA 1.5 13B (Liu et al., 2023a) and ShareCaptioner (Chen et al., 2023b).

Moreover, during the language data synthesis process in the DS stage, we also utilize language models (LM) to rewrite conversations using the provided tail tokens. The corresponding prompts are shown in Figure 10. Additionally, we rewrite conversations containing tail tokens or interrogation entities (TWR in the ablation study or Section 6.2). As this task closely resembles standard rephrasing tasks with similar prompts, we will not elaborate on it further here.

**Instruction:**
You are an AI visual assistant, and you are seeing a single image. What you see are provided with a few of sentences, describing the same image you are looking at. Answer all questions as you are seeing the image.
Design a conversation between you and a person asking about this photo. The answers should be in a tone that a visual AI assistant is seeing the image and answering the question.
Ask diverse questions and give corresponding answers.

Include questions asking about the visual content of the image, including the object types, counting the objects, object actions, object locations, relative positions between objects, etc. Only include questions that have definite answers:
(1) one can see the content in the image that the question asks about and can answer confidently;
(2) one can determine confidently from the image that it is not in the image.
Do not ask any question that cannot be answered confidently.

Also include complex questions that are relevant to the content in the image, for example, asking about background knowledge of the objects in the image, asking to discuss about events happening in the image, etc. Again, do not ask about uncertain details.
Provide detailed answers when answering complex questions. For example, give detailed examples or reasoning steps to make the content more convincing and well-organized. You can include multiple paragraphs if necessary.

**Fewshots:**
Caption:
There is a movie theater that displays the show times above the doors.
A red fire hydrant is deep in the snow.
The fire hydrant is in the snow near a recently plowed sidewalk.
This city has had a very hard winter with snow.
A hotel for dogs in the snow in winter.
Conversation:
Question: What color is the fire hydrant in the image?
Answer:The fire hydrant in the image is red.
Question:Is the fire hydrant fully covered by snow or partially visible?
Answer:The fire hydrant is partially visible, with a significant portion of it covered by the deep snow.
Question:What is the condition of the sidewalk near the fire hydrant?
Answer:The sidewalk near the fire hydrant has been recently plowed, so it is relatively clear of snow compared to the surrounding area.
Question:What does the image suggest about the winter season in the city?
Answer:The image suggests that the winter season in the city has been quite harsh and challenging. The significant snow accumulation seen in the image indicates that the city has experienced heavy snowfall, which can lead to various difficulties and disruptions for its residents.

For instance, the deep snow covering the fire hydrant could potentially hinder its accessibility in case of emergencies, which poses safety concerns. The recently plowed sidewalk implies that the city's maintenance crews have been actively working to keep the walkways clear and safe for pedestrians, but the sheer amount of snowfall might make it difficult to keep up with the continuous snow removal efforts.

Furthermore, such extreme winter conditions can impact transportation, leading to traffic issues, delays in public transit, and increased risks of accidents due to slippery roads. It can also cause problems for businesses and schools, as people might face challenges commuting to work or attending classes. Additionally, the heavy snow can put extra strain on infrastructure, such as roofs and power lines, increasing the likelihood of structural damage or power outages.

In conclusion, the image of the red fire hydrant deep in the snow and the recently plowed sidewalk suggest that the city has faced a particularly severe winter season, with substantial snowfall that has likely caused various challenges and disruptions for its residents and infrastructure.

Figure 9: Complete prompts used to guide the language model in converting captions into conversation instructions.

**Instruction:**
You are an AI language assistant involved in interpreting a conversation between a person and an AI visual assistant. The conversation revolves around an image. The task is to rephrase the conversation using a set of candidate words while maintaining the original meaning. The rephrased conversation must follow these criteria:
1. The conversation must remain coherent and grammatically correct.
2. One or more words from the 'Candidate words' list can be used to replace the original terms.
3. Words from the list can be used in any form (noun, verb, adjective, etc.), and each word may be used once or multiple times.
4. Not all words from the 'Candidate words' list need to be included; they are to be used based on context.
5. The conversation may be extended or shortened, but its meaning must remain unchanged.

**Fewshots:**
Conversation 1:
Question: What color is the traffic light shown in the image?
Answer: The traffic light in the image is green.
Question: How does the traffic appear to be moving at the intersection?
Answer: Traffic appears to be moving smoothly through the intersection, with cars continuing on their way after the green light.
Question: Is this image taken during the day or at night?
Answer: The image is taken at night.
Question: How are the traffic lights positioned in relation to the road?
Answer: The traffic lights are suspended above the road, hanging from a pole.
Question: How do the cars look in the image due to their motion?
Answer: Due to their motion, the cars appear as streaks passing by the traffic signals in the image. This effect indicates they are moving at moderate to high speeds while the image was captured.
===
Candidate words:
[overwinter, wintertime, set_phrase, give_voice, musical_phrase, phrasal_idiom, idiomatic_expression, articulate, formulate, idiom, get_behind, chase_after, drop_behind, hang_back, give_chase, go_after, drop_back, shack, twelvemonth, yr, railroad_car, elevator_car, cable_car, railway_car, motorcar, railcar, gondola, promiscuous, unaccented, light_up, abstemious, Light_Within, light_source, low-cal, Inner_Light, get_down, luminosity, wakeful, sluttish, luminousness, lightheaded, light-colored, fire_up, unclouded, visible_radiation, scant, visible_light, lightly, unhorse, light-headed, get_off, calorie-free, lightsome, swooning, illume, illumine, brightness_level, Christ_Within, ignitor, alight, wanton, weak, luminance, igniter, lighter, tripping, ignite, loose, faint, dismount, idle, illuminate, sparkle, twinkle, lightness, lite, easy, look-alike, figure_of_speech, simulacrum, range_of_a_function, mental_image, visualise, ikon, visualize, envision, effigy, trope, epitome, fancy, paradigm, see, word_of_honor, Holy_Writ, give_voice, countersign, Holy_Scripture, Good_Book, watchword, tidings, Christian_Bible, Word_of_God, give-and-take, articulate, Logos, formulate, parole, Son, Scripture, oppugn, interrogative_sentence, interrogate, dubiousness, doubtfulness, call_into_question, interrogative, interrogation, enquiry, fourth_dimension, prison_term, clock_time, metre]

Rephrased Conversation:
Question: What hue is the traffic signal displayed in the visual?
Answer: The traffic signal shown is illuminated in green.
Question: In what manner is the vehicular movement at the crossing?
Answer: Vehicular movement at the crossing is unobstructed, with motorcars proceeding post the green illumination.
Question: Was this visual captured during daylight or after dusk?
Answer: This visual was captured after dusk.
Question: In what relation are the traffic signals positioned to the roadway?
Answer: The traffic signals are suspended over the roadway, hanging from a pole.
Question: What appearance do the automobiles present in the visual due to their motion?
Answer: Owing to their motion, the automobiles are depicted as blurs traversing past the traffic signals, indicating their brisk pace at the time of capture.

Figure 10: Complete prompts used to guide the language model in rewrite conversation instructions using given tokens.

