# OpenReview forum: "Demystifying the Underappreciated Long-Tail Problems in Large Vision Language Models"
_ICLR.cc/2025/Conference — ICLR 2025 Conference Withdrawn Submission_

### Official Review · Reviewer_DBbC · 2024-10-27

**Soundness:** 3
**Presentation:** 2
**Contribution:** 2
**Rating:** 3
**Confidence:** 3

**Summary:**

This paper addresses the issue of imbalanced data distributions in training Large Vision-Language Models (LVLMs) by proposing an Adaptive Data Refinement Framework (ADR). The approach consists of two stages: Data Rebalancing (DR) and Data Synthesis (DS). In the DR stage, redundant and low-quality instances from overrepresented head data are filtered to create a more balanced dataset, enhancing the model's generalization capabilities. The DS stage enriches the training set by synthesizing new instances for scarce tail data using latent representations, adding diversity without increasing the overall data volume or training effort. Together, these stages aim to improve LVLM performance, particularly on tail concepts, by addressing data imbalance efficiently.

**Strengths:**

1. Balanced Data Representation: The ADR framework effectively filters redundant head data, ensuring more equitable representation of both head and tail instances, thus enhancing model generalization.
2. Tail Data Synthesis: By synthesizing new instances of underrepresented tail data, the framework enriches the training set, boosting performance on less frequent concepts.
3. Efficiency: ADR improves LVLM performance without increasing training data volume or requiring additional training, offering a resource-efficient solution to the long-tail problem.

**Weaknesses:**

1. Limited Impact of Certain Perspectives: The ablation study shows that some perspectives, such as co-occurrence and interrogation, do not significantly enhance performance, suggesting not all data perspectives are equally beneficial.
2. Limited Novelty: The approach largely relies on well-established data rebalancing and synthesis techniques based on object and token frequencies. While effective, it doesn't introduce fundamentally new concepts or methods, making the contribution incremental rather than groundbreaking.

**Questions:**

1. It appears that tokens and objects drive most of the performance gains. Given this, what is the rationale for including perspectives like interrogation and co-occurrence, which show limited impact?
2. If only the textual data were parsed for objects and tokens, without processing the images, how would the results be affected? It seems possible to achieve similar outcomes without image data, which raises questions about the necessity of visual inputs for the observed improvements.

---

### Official Review · Reviewer_E1cG · 2024-10-28

**Soundness:** 2
**Presentation:** 2
**Contribution:** 2
**Rating:** 5
**Confidence:** 4

**Summary:**

This paper proposes an approach to mitigate the long tail distribution problem in the training data of large vision-language models. The approach named Adaptive Data Refinement Framework (ADR) consists of two stages: Data Rebalancing (DR) and Data Synthesis (DS).
The approach is applied on the training data of LLaVA 1.5 and ShareGPT-4V separately. The new models trained on the rectified training data are evaluated across 10 benchmarks and demonstrate performance improvement over the original models.

**Strengths:**

1) This paper tackles an interesting and important problem - long-tail distribution in training data of large vision-language models.
2) The paper analyzes the long-tail distribution in the training data of LLaVA 1.5, and shows the correlation between prediction errors on benchmarks and the long-tail distribution
3) The proposed data refinement framework leads to performance improvement on LLaVA 1.5 and ShareGPT-4V

**Weaknesses:**

1) In the summary of contributions (line 91, line 96), the paper claims that the proposed framework ADR enhances the performance of large VLMs “without introducing any additional training”. This claim might be misleading as additional visual instruction tuning is required to train a new model on the data refined with ADR. The authors could consider clarifying or rephrasing this claim mentioning the required visual instruction tuning step.
2) In line 402, an off-the-shelf vision captioner is used to generate a caption for the synthetic image. Then the caption is extended into a conversation using a language model.
In Section C2, it is mentioned that LLaVA 1.5 13B and ShareCaptioner are used as the captioner, and LLaMA3 is used as the LM for generating the conversation. How dense are the captions generated by LLaVA 1.5 and ShareCaptioner? How to prevent hallucination when generating the conversation from the caption without passing the image as input?
The authors could provide details on  a) the average length or level of detail in the captions generated by LLaVA 1.5 and ShareCaptioner  b) techniques or safeguards used to ensure the generated conversations remain faithful to the original image content without introducing hallucinations.
3) Qualitative examples that show the quality of the generated data from the Data Synthesis step are missing. The authors could include a few representative examples of synthesized data showing the original image, generated caption, and resulting conversation.

**Questions:**

1) How to interpret the x-axis Q_r in Figure 4. Are these the indices of words that are ordered according to the occurrence frequency?

---

### Official Review · Reviewer_KfdZ · 2024-10-30

**Soundness:** 3
**Presentation:** 3
**Contribution:** 3
**Rating:** 6
**Confidence:** 5

**Summary:**

This paper focuses on the long-tail problem in MLLMs, which is critical for both academic research and practical application. The authors propose an adaptive data refinement framework to address this issue, which combine the strategies of data rebalancing and data synthesis. The experimental results well validate the effectiveness of this approach.

**Strengths:**

1. The studied problem of this paper, i.e., long-tail data distribution, is important and significant in existing MLLMs.

2. The proposed ADR can alleviate this issue to some extend even by simply using data rebalancing without new training data.

3. The overall presentation is good, and the paper is easy to follow.

**Weaknesses:**

1. The novelty of the proposed ADR method is not well described. Rebalancing and data synthesis are common solutions for long-tail problem, but what are the main differences and advantages of the proposed methods, it is not very clearly stated. Besides, it is better to introduce the principle and methodology of the compared data-rebalancing methods for comparison.

2. The effectiveness of ADR's rebalancing scheme is obvious compared to the default LLaVA SFT data scheme, but its advantages seem marginal compared to the rebalancing baselines. In tab.5, ADR's rebalancing is slightly better than perplexity but close to Random, especially for the long-tail task VizWiz. The authors are expected to answer and analyze this case.

Minor:

I would like to suggest the authors to gain more in-depth insights into the long-tail problems of MLLMs. In addition to performance improvement, what insights can we obtain from the long-tail study? For instance, visual hallucination is often regarded as the main problem of MLLMs, and most people think that it is related to the visual feature quality. But in early VL study like VQA, visual hallucination is also related to language bias, i.e., the model guess the answer according to the data distribution. In some case, it is also a problem of long-tail data distribution.

So, it would be bette if we can see that the ADR can solve some important problems of MLLMs in addition to performance.

**Questions:**

Most of my concerns and questions are given in the weakness.

---

### Official Review · Reviewer_Q7sJ · 2024-11-02

**Soundness:** 3
**Presentation:** 3
**Contribution:** 2
**Rating:** 3
**Confidence:** 5

**Summary:**

This paper presents a comprehensive analysis of the long-tail problem in the instruction tuning data of LVLM from four perspectives: token, object, co-occurrence, and interrogation. Based on this analysis, the authors propose an Adaptive Data Refinement Framework, which consists of two stages: Data Rebalancing and Data Synthesis. To validate the framework's effectiveness, the authors conduct extensive experiments on 10 benchmarks.

**Strengths:**

1.	The experiments are sufficient. The authors conducted experiments on multiple benchmarks and on different baselines, such as LLaVA 1.5 and ShareGPT4V.
2.	The writing is fluent and easy to understand.

**Weaknesses:**

1.	The significance of addressing the long-tail problem during the instruction-tuning stage remains ambiguous.
During instruction-tuning stage, the LLM is usually trainable, leading to a large scale of trainable parameters, therefore, LVLM can be easily fitting to the training data. A typical phenomenon is that the loss will suddenly decrease after the first epoch. Given this, the long-tail problem is unlikely to be a pivotal concern during the instruction-tuning stage for LVLM. Instead, the long-tail problem may assume greater importance during the pretraining stage for LVLM.
2.	Absence of visualization results.
As Data Synthesis Stage is an important part of the proposed method, the authors should provide examples of synthetic data, including images and conversations.
3.	The impact of the Data Balance Stage is inconspicuous.
As shown in Tab 5.,  when compared to randomly sampled training data, the performance of ‘our-balance’ does not show a significant improvement. In contrast, it even declines in some benchmarks, such as VizWiz, TextVQA, and MMstar. Based on the comparisons in Table 5, the improvement attributed to the Data Balance Stage is unlikely due to the balanced nature of the training data, but rather the reduction in training data redundancy.
4.	Insufficient discussion of related works.
There exist other studies that explore leveraging synthetic data to tackle the long-tail problem. The authors should discuss the difference between the proposed method and these related works[1][2].
[1]	LTGC: Long-tail Recognition via Leveraging LLMs-driven Generated Content
[2]	Balancing the Picture: Debiasing Vision-Language Datasets with Synthetic Contrast Sets

**Questions:**

1.	It is unclear why addressing the long-tail (LT) problem in instruction-tuning data would be beneficial for test benchmarks.
As the authors mention, the distribution of training and test data differs. I am curious about why tackling the LT problem in instruction-tuning data is advantageous for test benchmarks. The test benchmarks can be considered as extreme long-tail cases within the training data, occurring zero times during training. From this perspective, the long-tail issue in test benchmarks still persists.
2.	What is the performance of synthesizing data with the balance strategy of random sampling?
3.	What is the performance when only the Data Synthesis Stage is implemented?
4.	How can the author ensure that the synthetic data is correct? Hallucinations in captions may lead to errors in the conversations generated by LLMs.

---

### Official Review · Reviewer_HeWS · 2024-11-03

**Soundness:** 2
**Presentation:** 2
**Contribution:** 2
**Rating:** 5
**Confidence:** 4

**Summary:**

In this work, the authors present an approach to tackle the long-tail distribution problem present in the training data of modern vlms. They have two steps of the methodology which first balances the data in the long tail and then second can generate the data to mitigate the long tail. Experiments are performed on a two VLMs and they show some improvements.

**Strengths:**

1. The proposed approach tackles a relevant problem, at least for some of the present vlms, however, questions still remain over the overall usefulness.
2. The method involves two parts to solve the problem. Which seems logical.
3. The experiments show some improvements, however, minor.

**Weaknesses:**

1. The paper can be written more clearly. It is quite dense in some parts, such as the main method section.
2. The experiments are performed on two vlms. Due to the rapidly evolving landscape. I find these experiments a little less. Maybe it could've been more useful to experiment with other vlms as well.
3. The section 3.4 has some trivial findings. All three findings are pretty general and well-known to the community. I am wondering about the usefulness of this section.
4. I am also wondering about the overall motivation of the paper. More specifically, wouldn't scale just solve the problem of long-tail distributions in the data?

**Questions:**

I have a few questions:

1. Importantly, with the ever-growing interest of the community in VLMs. Is this problem really important? For example, in the recent 'Molmo and PixMo' paper - they proposed a large-scale dataset. Would the scale automatically solve this problem?
2. Can the authors also report results with more datasets?
3. I am wondering about some of the choices in the 'Entity Distribution Construction' - did the authors try some other ways? How did they choose these particular ways?
4. Interrogation: how did the authors choose the categories of these questions in the data? Is there some existing taxonomy which they followed?

---

### Note · Authors · 2024-11-14

**Comment:**

After further review of our work, we have identified areas that require significant improvement. To ensure the accuracy and quality of our research, we have decided to withdraw our submission. We sincerely thank the reviewers for their valuable feedback, which has provided us with insightful suggestions and helped us identify key areas for improvement. We appreciate your understanding.

**Withdrawal Confirmation:**

I have read and agree with the venue's withdrawal policy on behalf of myself and my co-authors.